# Effect of Androgen-Deprivation Therapy on Bone Mineral Density in Patients with Prostate Cancer: A Systematic Review and Meta-Analysis

**DOI:** 10.3390/jcm8010113

**Published:** 2019-01-18

**Authors:** Do Kyung Kim, Joo Yong Lee, Kwang Joon Kim, Namki Hong, Jong Won Kim, Yoon Soo Hah, Kyo Chul Koo, Jae Heon Kim, Kang Su Cho

**Affiliations:** 1Department of Urology, Gangnam Severance Hospital, Urological Science Institute, Yonsei University College of Medicine, Seoul 06273, Korea; dokyung@yuhs.ac (D.K.K.); UROHAH@yuhs.ac (Y.S.H.); GCKOO@yuhs.ac (K.C.K.); 2Department of Urology, Severance Hospital, Urological Science Institute, Yonsei University College of Medicine, Seoul 03722, Korea; JOOURO@yuhs.ac (J.Y.L.); DOCTOR2PLAY@yuhs.ac (J.W.K.); 3Division of Geriatrics, Department of Internal Medicine, Severance Hospital, Yonsei University College of Medicine, Seoul 03722, Korea; PREPPIE@yuhs.ac; 4Department of Internal Medicine, Severance Hospital, Endocrine Research Institute, Yonsei University College of Medicine, Seoul 03722, Korea; NKHONG84@yuhs.ac; 5Department of Urology, Soonchunhyang University Hospital, Soonchunhyang University College of Medicine, Seoul 04401, Korea; piacekjh@hanmail.net

**Keywords:** androgen deprivation therapy, bone mineral density, prostate cancer, systematic review, meta-analysis

## Abstract

We aimed to evaluate the change in bone mineral density (BMD) in patients with prostate cancer (PCa) receiving androgen deprivation therapy (ADT) compared to those with PCa or other urologic conditions not receiving ADT. Literature searches were conducted throughout October 2018. The eligibility of each study was assessed according to Preferred Reporting Items for Systematic Reviews and Meta-Analyses guidelines using the Participant, Intervention, Comparator, Outcome, and Study design method. The outcomes analyzed were the mean difference (MD) of percent changes in BMD of lumbar spine, femur neck, and total hip. Five prospective cohort studies with a total of 533 patients were included in the present study. Statistically significant decreases of BMD change relative to the control group were observed in the ADT treatment group in the lumbar spine (MD −3.60, 95% CI −6.72 to −0.47, *P* = 0.02), femoral neck (MD −3.11, 95% CI −4.73 to −1.48, *P* = 0.0002), and total hip (MD −1.59, 95% CI −2.99 to −0.19, *P* = 0.03). There is a significant relationship between ADT and BMD reduction in patients with PCa. Regular BMD testing and the optimal treatment for BMD loss should, therefore, be considered in patients with PCa undergoing ADT.

## 1. Introduction

Prostate cancer (PCa) is the most common malignancy among men [1]. Improved screening and management of the disease have led to earlier diagnosis and longer life expectancy for patients. Due to the many side effects associated with treatment options, the quality of life for these patients is becoming increasingly important [2]. Despite local treatment, the natural course of PCa in 40% of patients is metastasis, especially in the bone [2]. Bone metastasis contributes to mortality and is the major cause of morbidity due to skeletal-related events, including fractures and spinal cord compressions, and the need for surgery or radiation therapy as therapeutic or palliative measures [3].

PCa is an androgen-dependent disease, and androgen deprivation therapy (ADT) is the mainstay of treatment for hormone-sensitive metastatic or advanced PCa [4]. In castrate-resistant prostate cancer (CRPC), docetaxel chemotherapy and second-line hormone treatments, such as abiraterone or enzalutamide, have been introduced. However, ADT must be maintained in CRPC [5]. Furthermore, patients with clinically localized PCa are usually treated with radical prostatectomy (RP), radiation therapy (RT), or active surveillance [6]. Disease recurrence most often manifests as an increase in prostate-specific antigen, so salvage therapy (RP, RT, cryoablation, high-intensity focused ultrasound (HIFU)), and androgen deprivation therapy (ADT) were commonly applied in local recurrence of PCa [7]. 

ADT includes induction of hypogonadism through orchiectomy and a luteinizing hormone-releasing hormone (LH-RH) agonist, alone or combined with an androgen blockade (LH-RH agonist plus antiandrogen) [8]. Although ADT is highly effective, it can result in many problematic complications related to long-term use, including osteoporosis with reduced bone mineral density (BMD) [9,10,11]. Undoubtedly, bone health is an important concern for patients with PCa. BMD may decrease by up to 13% yearly in men receiving ADT [12]. Moreover, men with PCa may also experience significant bone loss due to disease, even before the induction of ADT [6]. Since many patients with PCa tend to be older, BMD loss is superimposed on the gradual decrease in bone density that accompanies normal aging [13]. Cumulative decrease in BMD is related to an increased risk of fracture [14], which can increase morbidity and mortality [15]. Diagnosed patients are susceptible to osteoporosis according to their age, but most are receiving ADT [2]. These factors emphasize the fact that reduction in BMD associated with ADT is becoming increasingly prevalent and important in patients with PCa. 

Although numerous research studies have been conducted on the relationship between the use of ADT and BMD reduction [9,10,11,16,17,18,19,20,21,22], there has not been a systematic review and meta-analysis of this topic in existing literature. Therefore, we conducted this study as a systematic review of published literature and meta-analysis of available data in order to evaluate the change in BMD in patients with PCa receiving ADT compared to those who did not receive ADT.

## 2. Materials and Methods

This systematic review was registered in PROSPERO (CRD 42018107948). 

### 2.1. Search Strategy

Computerized bibliographic search of PubMed or Medline, Embase, and Cochrane Library databases was conducted through October 2018. Search terms included “prostate cancer”, “androgen deprivation OR androgen suppression OR hormone OR gonadotropin” and “bone mineral density OR bone loss OR bone density OR skeletal change OR osteoporosis”. Search terms used for PubMed or Medline and Embase are listed in the Appendix A. Conference and meeting abstracts were excluded, even if they met the eligibility criteria. In the end, our search identified 482 candidate articles. Two authors (DKK and YSH) independently reviewed the titles and abstracts according to our inclusion and exclusion criteria, and subsequently reviewed the identified articles.

### 2.2. Trial Inclusion and Exclusion Criteria

We assessed the eligibility of each study according to Preferred Reporting Items for Systematic Reviews and Meta-Analyses (PRISMA) guidelines using the Participant, Intervention, Comparator, Outcome, and Study design (PICOS) method [23].

Study population was defined as patients with PCa who were treated with ADT. Patients with PCa or other urologic conditions (e.g., benign prostatic hyperplasia, urologic stone, or erectile dysfunction) who were not treated with ADT were defined as the comparator. The analyzed outcomes included percent changes in BMD of lumbar spine, femur neck, and total hip. Inclusion criteria included a study published in English, prospective cohort design, patients with PCa or other urologic diseases, use of ADT, ADT duration of at least 6 months, follow-up period of at least 1 year, and reported values for changes in BMD of lumbar spine, femur neck, or total hip. Exclusion criteria included a cohort observational study design (no control group), use of intermittent ADT regimen, a comparison between regimens of ADT, short follow-up period (< 12 months), and inability to extract outcome data. Conference and meeting abstract was also excluded.

### 2.3. Data Extraction

Two authors (DKK and YSH) reviewed all of the included articles and independently extracted data from each study. Any discrepancies between the two authors in extracted data were resolved via consensus. Extracted data included study design details, inclusion and exclusion criteria, participant demographics, treatment characteristics (regimen, dosage, and duration), measured outcomes (BMD of lumbar spine, femur neck, and total hip), and results (percent change of BMD, mean difference (MD), and standard deviation (SD)).

### 2.4. Study Quality Assessments and Quality of Evidence

The quality of included clinical trials was evaluated according to the methodological index of Downs and Black scale. This index is comprised of five major assessment categories, including reporting, external validity, bias, confounding, and power [24]. 

Grading of Recommendations, Assessments, Developments, and Evaluation (GRADE) system provided a systematic approach for evaluating the quality of evidence and strength of recommendations [25]. The certainty of comparisons was evaluated with GRADE system, using assessments of the following criteria: methodology, precision, consistency, directness, and risk of publication bias. Based on these criteria, we assessed evidence of comparisons by classifying the quality of evidence on a four-level scale (i.e., high, moderate, low, and very low). 

### 2.5. Statistical Analysis

Percent changes in BMD outcomes were measured and recorded as continuous data. Values of MD and SD were extracted from all studies. The pooled MD for ADT and control group values and 95% CIs were calculated. Meta-analyses were performed using the random-effects model of DerSimonian and Laird to obtain pooled overall MD with 95% CIs for outcomes [26].

Statistical heterogeneity was assessed using I^2^ value and χ^2^ test. A Cochran Q statistic of *P* < 0.05 or I^2^ > 50% indicated the presence of statistically significant heterogeneity. 

Meta-analysis was performed using Review Manager v.5.1 (Nordic Cochrane Center, Cochrane Collaboration, Copenhagen, Denmark, 2008). All P-values were two-sided, and except for the test of discrepancy, a *P* < 0.05 was considered to indicate a statistically significant result. Since fewer than 10 studies were included in our study, we did not follow through with a plan to use funnel plots to assess small study effects. 

## 3. Results

### 3.1. Systematic Review Process

We used PRISMA statements to analyze and summarize our systematic analysis and meta-review process (Figure 1). Only published studies were included to minimize publication bias. Initial database searches identified 2442 articles, which were reduced to 1778 following duplicate removal. Subsequently, 1738 articles were removed after review of title and abstract. Analysis of the remaining full-text articles, with respect to inclusion and exclusion criteria, resulted in the final selection of five studies with a total of 533 patients (Table 1). All included studies were prospective cohort studies. The majority of the study population were clinically localized or advanced PCa patients who underwent ADT after diagnosis. The ADT in these studies included bilateral orchiectomy, LH-RH agonist, and anti-androgen medication, and the duration of ADT ranged from 6 to 36 months. The duration of follow-up ranged from 1 to 3 years. Excluded studies are listed in the Appendix A.

### 3.2. Outcome Comparisons between ADT and Control Groups

#### 3.2.1. Lumbar Spine: Percent Change of BMD

Analysis of the percent change of BMD in lumbar spine included four studies with 483 patients (Figure 2A). A statistically significant decrease of lumbar spine BMD change was observed in ADT group relative to control group (MD −3.60, 95% CI −6.72 to −0.47, *P* = 0.02). Between-study heterogeneity was observed (I^2^ = 99%, *P* < 0.00001).

#### 3.2.2. Femoral Neck: Percent Change of BMD

Analysis of the percent change of BMD in femoral neck included five studies with 515 patients (Figure 2B). A statistically significant decrease of femoral neck BMD change was observed in ADT group relative to control group (MD −3.11, 95% CI −4.73 to −1.48, *P* = 0.0002). Between-study heterogeneity was observed (I^2^ = 82%, *P* = 0.0002).

#### 3.2.3. Total Hip: Percent Change of BMD

Analysis of the percent change of BMD in total hip included four studies with 483 patients (Figure 2C). A statistically significant decrease of BMD change of total hip was observed following ADT treatment relative to control group (MD −1.59, 95% CI −2.99 to −0.19, *P* = 0.03). Between-study heterogeneity was observed (I^2^ = 90%, *P* < 0.00001).

### 3.3. Quality Assessment and Qualitative Risk of Bias

Downs and Black scale was utilized to assess the quality of five prospective trials using reporting, external validity, bias, confounding, and power assessment categories (Table 2). Downs and Black scores of the evaluated studies ranged from 13 to 15. The results of GRADE quality assessment of direct evidence of each comparison are shown in Table 3. Certainty was “low” in all three comparisons.

## 4. Discussion

Healthy bone is in equilibrium with ongoing bone formation and bone resorption, which is normally mediated by osteoblasts and osteoclasts [6]. Hormones, such as estrogens and androgens, help balance this equilibrium between bone synthesis and degradation [27]. However, this equilibrium is unbalanced in severely hypogonadal men, who experience decreased BMD and severe bone architecture damage [28]. Unfortunately, ADT for PCa patients interferes with the normal hormonal balance needed for bone health. The rate of BMD loss that occurs in patients receiving ADT is significantly higher than that caused by normal aging or female menopause. Men experiencing normal aging lose BMD at a rate of approximately 0.5% to 1.0% yearly until middle age. Women experiencing normal aging lose bone mass at a similar rate until menopause, and then the rate of bone density decline increases every year for 5 years (approximately 3% yearly in the spine). Bone loss associated with ADT is more rapid and severe than that in normal aging men or women [6]. For example, the bone loss rates in the lumbar spine and femoral neck regions of PCa patients after initiation of treatment with ADT have been reported as 4.6% and 3.9% [17]. Numerous prospective studies have documented the substantial bone loss that occurs in men with PCa who are treated with ADT [9,10,11,16,17,18,19,20,21,22]. To better understand the findings of these studies, we examined the effects of ADT on BMD in PCa patients through systematic review and meta-analysis of prospective cohort studies. 

Our meta-analysis discerned a significant decline in BMD at the lumbar spine, femoral neck, and total hip regions of PCa patients treated with ADT compared to controls. ADT causes a decrease in BMD by affecting both the trabecular and cortical bones [29,30]. Bone loss due to ADT increases the risk of fracture exponentially. Shahinian et al. [14] showed that among men surviving more than 5 years after diagnosis of PCa, bone fractures were noted in about 20% of those who received ADT, compared with about 10% of those who did not receive ADT (*P* < 0.001). These authors also reported a significant relationship between the number of doses of LH-RH agonist administered during the first year and fracture risk. Our analysis demonstrated a greater effect of ADT on BMD in the lumbar spine than in the femoral neck or total hip, which is consistent with several other studies in men who received ADT, in which decreases in lumbar spine BMD are greater than other measured areas [20,31,32]. Lumbar spine has a higher percentage of trabecular bone than total hip or femoral neck. Since trabecular bone is metabolically more active than cortical bone, it may be more sensitive to ADT [19,33]. Moreover, lumbar spine is the most common site of fractures due to ADT, and BMD reduction by ADT may be a predisposing factor to vertebral compression fractures [14,34]. 

Although it is clear that early detection of bone loss and prompt initiation of preventive and pharmacological measures to delay or prevent decreased BMD are essential to reduce the risk of bone fracture for advanced PCa patients who have begun ADT, clinical practice guidelines on BMD evaluation in patients with PCa on ADT are not clear-cut [35]. Some suggestions have addressed this issue. Diamond et al. [36] proposed that BMD should be assessed in patients considered to be at high risk for osteoporosis and all men who have a risk factor for fracture, like those receiving ADT or with a history of fracture. The USA Endocrine Society and the National Comprehensive Cancer Network have proposed to measure BMD in men aged 50–69 years with risk factors (e.g., ADT) [33,37]. National Comprehensive Cancer Network guidelines regarding ADT-induced bone loss also recommend supplementation with either 60 mg of denosumab subcutaneously every 6 months, 5 mg of zoledronic acid intravenously every year, or 70 mg of alendronate orally every week for men with a 10-year risk of hip fracture greater than 3%, calculated using the fracture risk assessment tool [37,38]. Treatments for ADT-induced BMD loss include bisphosphonates, human monoclonal antibody (denosumab), and selective estrogen receptor modulators (e.g., raloxifene and toremifene) [39]. There have been many randomized controlled trials (RCTs) about the efficacy of these osteoporotic medications for bone loss of PCa patients due to ADT. A systematic review and network meta-analysis to compare the effectiveness of various osteoporotic treatments (bisphosphonates, denosumab, toremifene, and raloxifene) on BMD loss in patients with non-metastatic PCa on ADT, performed by Poon et al., [39], found that that all drugs are effective in reducing the rate of bone loss, but did not find evidence that one drug is more effective than another. Lifestyle modifications and nutritional supplementation can have a significant effect on bone health, and may delay the onset and severity of ADT-related bone loss. Regular exercise reduces the risk of fractures by reducing bone loss, increasing bone and muscle strength, and improving mobility [40]. Nutritional intervention is a simple way to determine whether a patient is receiving adequate levels of minerals and vitamins, especially calcium and vitamin D, to maintain proper bone formation. Therefore, clinicians administering ADT for PCa patients should always be mindful of periodic BMD testing. Moreover, they should encourage lifestyle interventions and nutritional supplements, provide some form of osteoporosis treatment to men who are at risk of fracture, and choose the optimal drug based on efficacy, safety, patient preferences, patient and health system costs, and local availability [39].

Many meta-analyses have shown the effects of ADT on cognitive function, cardiovascular complications, and thromboembolic events in PCa patients [41,42,43,44,45]; however, to the best of our knowledge, our study is the first to show the effect of ADT on bone health in patients with PCa. Furthermore, we have identified a significant relationship between ADT and BMD reduction. The present study analyzed only prospective cohort studies, and the level of evidence was lower than that of meta-analyses with only RCTs (level of evidence was evaluated using GRADE, and certainty was low in all cases). Although only prospective cohort studies were included and inconsistency was serious for all evidence, the level of evidence was elevated to “low” from “very low”, due to a strong correlation between the duration of exposure to ADT and the decrease in BMD [46]. Control group of the studies included in this analysis was heterogeneous. Control group included patients with PCa, other urologic problems, or both. Other limitations of our analysis were the small number of included studies and the inconsistency in the follow-up period of included studies.

## 5. Conclusions

There is a significant relationship between ADT and BMD reduction in patients with PCa. Therefore, regular BMD testing should be considered in patients with PCa undergoing ADT. Moreover, determination of the optimal treatment, such as medical therapy, lifestyle intervention, and nutritional support for BMD loss based on various factors, should help identify patients who are at risk of fracture. Well-designed prospective RCTs are required to overcome the limitations of this study, as well as to establish evidence to support the results of the present study.

## Figures and Tables

**Figure 1 jcm-08-00113-f001:**
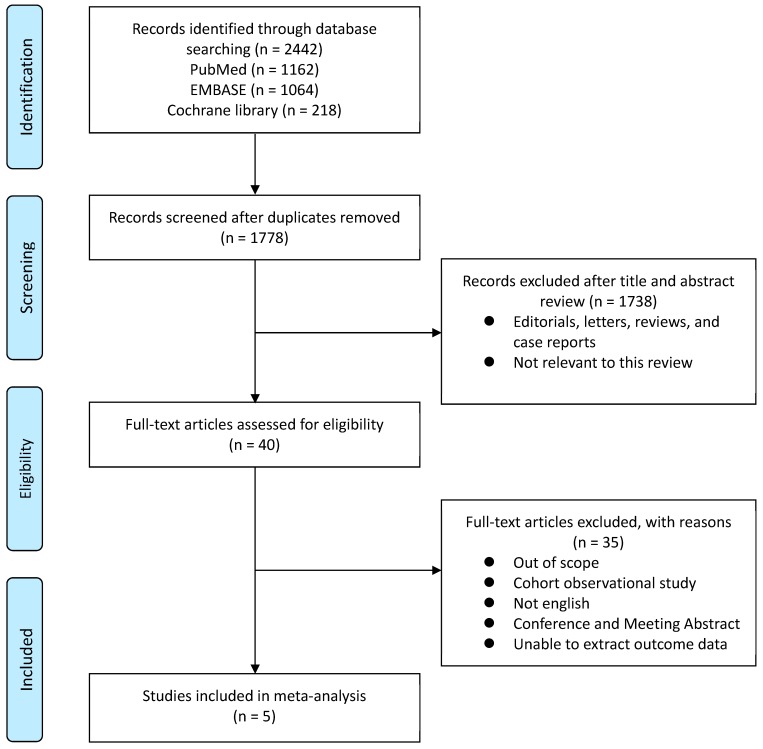
Preferred reporting items for systematic reviews and meta-analysis (PRISMA) flowchart. This chart shows the flow of information through different phases of systematic review and the exclusion criteria used.

**Figure 2 jcm-08-00113-f002:**
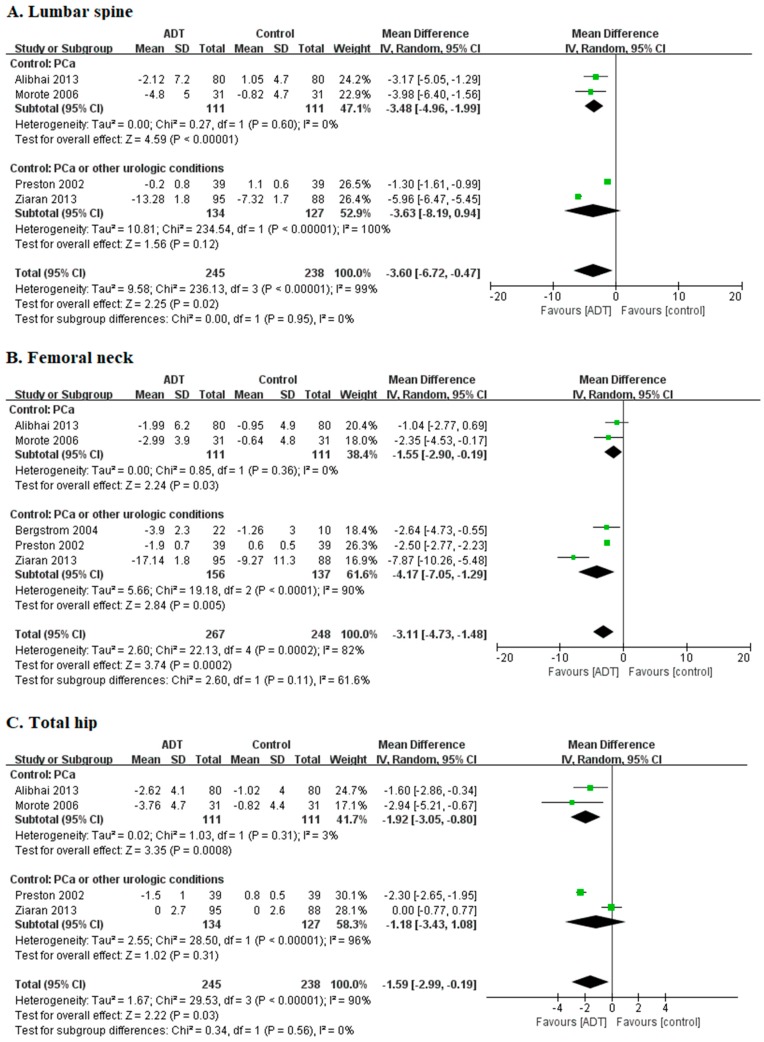
Forest plots comparing ADT and control groups. (**A**) Lumbar spine. (**B**) Femoral neck. (**C**) Total hip. ADT, androgen deprivation therapy; PCa, prostate cancer; Green box, weight; Central line of diamond, mean difference; Lateral tips of diamond, confidence intervals.

**Table 1 jcm-08-00113-t001:** Characteristics of eligible prospective cohort studies.

Study	Design	Group Characteristics (Total Number)		Tumor Stage (Total Number)	Duration of ADT	Follow-Up Period	BMD Check Site	BMD Change Outcome (SD)		Conflict of Interest
Alibhai et al. [19]	prospective cohort study	ADT	Patients with PCa who underwent continuous ADT for at least 1 year (80)	cT1c N0 M0 (22) cT2 N0 M0 (41) cT3 N0 M0 (17)	12–36 months	3 years	1. Lumbar spine 2. Femoral neck 3. Total hip	Lumbar spine	ADT: −2.12% (7.2)	None
									Control: −1.05% (4.7)	
								Femoral neck	ADT: −1.99% (6.2)	
		Control	Patients with PCa who were not on ADT (80)	cT1c N0 M0 (35) cT2 N0 M0 (43) cT3 N0 M0 (2)					Control: −0.95% (4.9)	
								Total hip	ADT: −2.62% (4.1)	
									Control: 1.02% (4.0)	
Bergstrom et al. [9]	prospective cohort study	ADT	Patients with either advanced PCa or recurrent disease following primary, local therapy who were treated with bilateral orchidectomy and GnRH analogues continuously (22)	NA	12 months	1 year	Femoral neck	ADT	−3.9% (2.3)	Stiftelsen Johanna Hagstrand och Sigfrid Linne’rs Minne and Karolinska Institutet Research funds
		Control	Patients with other urologic conditions such as BPH, stones (40)	NA				Control	−1.26% (3)	
Morote et al. [20]	prospective cohort study	ADT	Patients with PCa who underwent continuous ADT with 3 months of depot LH-RH agonist (31)	cT3a N0 M0 (14) cT3b-4 N0 M0 (7) cT2-4 N1 M0 (10)	12 months	1 year	1. Lumbar spine 2. Femoral neck 3. Total hip	Lumbar spine	ADT: −4.8% (5)	None
									Control: −0.82% (4.7)	
								Femoral neck	ADT: −2.99% (3.9)	
		Control	Patients with PCa free of BCR after RP (31)	cT1c N0 M0 (20) cT2a N0 M0 (11)					Control: −0.64% (4.8)	
								Total hip	ADT: −3.76% (4.7)	
									Control: −0.82% (4.4)	
Preston et al. [21]	prospective cohort study	ADT	Patients with PCa who had received continuous ADT for a minimum of 6 months for either advanced PCa on presentation or for recurrent disease following primary local therapy (RP or RT) (39)	NA	≥6 months	2 years	1. Lumbar spine 2. Femoral neck 3. Total hip	Lumbar spine	ADT: −0.2% (0.8)	U.S. Army Medical Research and Development Command
									Control: 1.1% (0.6)	
								Femoral neck	ADT: −1.9% (0.7)	
		Control	Patients with other urologic conditions, such as ED or BPH, and those with PCa who had completed primary therapy (RP or RT) with no evidence of disease (39)	NA					Control: 0.6% (0.5)	
								Total hip	ADT: −1.5% (1)	
									Control: −0.8% (0.5)	
Ziaran et al. [22]	prospective cohort study	ADT	Patients with locally advanced PCa (95)	cT3a N0 M0 (89) pT3b N0 M0 (6)	24 months	2 years	1. Lumbar spine 2. Femoral neck 3. Total hip	Lumbar spine	ADT: −13.28% (1.8)	None
									Control: −7.32% (1.7)	
								Femoral neck	ADT: −17.14% (1.8)	
		Control	Patients with other urologic conditions such as LUTS, stones, etc. (88)	NA					Control: −9.27% (11.3)	
								Total hip	ADT: 0% (2.7)	
									Control: 0% (2.6)	

ADT, androgen deprivation therapy; BCR, biochemical recurrence; BMD, bone mineral density; BPH, benign prostate hyperplasia; ED, erectile dysfunction; LH-RH, luteinizing hormone-releasing hormone; LUTS, lower urinary tract symptoms; NA, not available; PCa, prostate cancer; RP, radical prostatectomy; RT, radiation therapy; SD, standard deviation.

**Table 2 jcm-08-00113-t002:** Downs and Black scale for quality assessment.

	Reporting	External Validity	Internal Validity		Power	Total
			Bias	Confounding (Selection Bias)		
Alibhai et al. [19]	7	1	3	3	1	15
Bergstrom et al. [9]	6	1	3	2	1	13
Morote et al. [20]	7	1	3	3	1	15
Preston et al. [21]	7	1	3	4	1	15
Ziaran et al. [22]	6	1	3	4	1	14

**Table 3 jcm-08-00113-t003:** Grading of Recommendations, Assessments, Developments, and Evaluation (GRADE) quality assessment for direct evidence of each comparison.

Certainty Assessment							Number of Patients		Effect	Certainty	Importance
Number of Studies	Study Design	Risk of Bias	Inconsistency	Indirectness	Imprecision	Other Considerations	ADT	Control	Absolute (95% CI)		
Lumbar spine											
4	Prospective, cohort studies	Not serious	Serious ^a^	Not serious	Not serious	Dose–response gradient	245	238	MD 3.6 lower (6.72 lower to 0.47 lower)	●●◯◯ LOW	CRITICAL
Femoral neck											
5	Prospective, cohort studies	Not serious	Serious ^a^	Not serious	Not serious	Dose–response gradient	267	248	MD 3.11 lower (4.73 lower to 1.48 lower)	●●◯◯ LOW	CRITICAL
Total hip											
4	Prospective, cohort studies	Not serious	Serious ^a^	Not serious	Not serious	Dose–response gradient	245	238	MD 1.59 lower (2.99 lower to 0.19 lower)	●●◯◯ LOW	CRITICAL

ADT, androgen deprivation therapy; CI, confidence interval; MD, mean difference; ^a^ significant heterogeneity observed. LOW level of certainty means that further research is very likely to have an important impact on our confidence in the estimate of effect and is likely to change the estimate.

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
