# Peer review of "Effect of Androgen-Deprivation Therapy on Bone Mineral Density in Patients with Prostate Cancer: A Systematic Review and Meta-Analysis"

_jcm, 2019, doi:10.3390/jcm8010113_

Reviewer 1 Report

This is a well writing meta-analysis regarding the ADT on bone mineral density in prostate cancer patients. 

Some minor comments:

1: the study population, what's the clinical stages of prostate cancer patients receive ADT in this manuscript? majority with M1 disease ( with bone mets at initial diagnosis) or disease recurrence after Radical Prostatectomy/Radiotherapy?

2: the duration of ADT need clarified, like more than 9 months, one years ...etc., was into meta-analysis

   also, the duration of ADT shorter than ? months , should be excluded

Author Response

1. the study population, what's the clinical stages of prostate cancer patients receive ADT in this manuscript? majority with M1 disease (with bone mets at initial diagnosis) or disease recurrence after Radical Prostatectomy/Radiotherapy?

 [Answer] For the tumor stage of the study population, it was added to table 1. Most of them were clinically localized or advanced prostate cancer patients who underwent ADT after diagnosis. The disease recurrence after Radical Prostatectomy was about 6 patients. we revised the manuscript and table 1 as follows. All modifications are highlighted in the manuscript.

In 3.1. Systematic review process

The majority of the study population were clinically localized or advanced PCa patients who underwent ADT after diagnosis. The disease recurrence after RP was about 6 patients.

2. the duration of ADT need clarified, like more than 9 months, one years ...etc., was into meta-analysis

also, the duration of ADT shorter than ? months , should be excluded

[Answer] According to your comments, we clarified the duration of ADT treatment. we revised the manuscript and added the duration of ADT treatment to table 1 as follows. All modifications are highlighted in the manuscript.

In 3.1. Systematic review process

The ADT in these studies included bilateral orchiectomy, LH-RH agonist, and anti-androgen medication, and the duration of ADT ranged from 6 to 36 months. The duration of follow-up ranged from 1 to 3 years.

Reviewer 2 Report

This Meta-Analysis is well-written. The Topic is interesting and important in clinical setting.

The paper Needs minor revisions. I have some remarks:

 Introduction: page1/line 39 citation is missing. 

page 2/line 43: please include Active S. as an option.

page2/line 45-46: PSA elevation after RPX or radiation therapy is not in every case a metastatic desease. so you should include salvage therapy in local recurrence of PCa.

In the following Paragraph i miss new standards like early chemotherapy (doxetacel) or abiraterone.  

Results: page 3/line 120 please check 1739 articles in comparison to fig.1 (1738).

Conclusion: page5/line 226: which optimal treatment? more concrete.

Author Response

1. Introduction: page1/line 39 citation is missing.

[Answer] According to your comments, we have added a citation on page 1/line 39. All modifications are highlighted in the manuscript.

2. page 2/line 43: please include Active S. as an option.

[Answer] As you suggested, we included active surveillance as a treatment option for clinically localized prostate cancer. we revised the manuscript as follows. All modifications are highlighted in the manuscript.

In 1. Introduction

Furthermore, patients with clinically localized PCa are usually treated with radical prostatectomy (RP), radiation therapy (RT), or active surveillance [6].

3. page2/line 45-46: PSA elevation after RPX or radiation therapy is not in every case a metastatic disease. so you should include salvage therapy in local recurrence of PCa.

[Answer] As you suggested, we included salvage therapy in local recurrence of prostate cancer. We revised the manuscript as follows. All modifications are highlighted in the manuscript.

In 1. Introduction

Disease recurrences most often manifest as an increase in prostate-specific antigen, so salvage therapy (RP, RT, cryoablation, high intensity focused ultrasound [HIFU]) and ADT is commonly applied in local recurrence of PCa [7].

ADT includes induction of hypogonadism through orchiectomy and a luteinizing hormone-releasing hormone (LH-RH) agonist alone or combined with an androgen blockade (LH-RH agonist plus antiandrogen) [8].

4. In the following Paragraph i miss new standards like early chemotherapy (doxetacel) or abiraterone. 

[Answer] As you suggested, we included new standards like early chemotherapy (doxetacel) or abiraterone. We revised the manuscript as follows. All modifications are highlighted in the manuscript.

 In 1. Introduction

PCa is an androgen dependent disease and androgen deprivation therapy (ADT) is the mainstay of treatment of hormone sensitive metastatic or advanced prostate cancer [4]. In the castrate-resistant prostate cancer (CRPC), docetaxel chemotherapy and second line hormone treatments such as abiraterone or enzalutamide have been introduced, but ADT must be maintained in CRPC [5].

5. Results: page 3/line 120 please check 1739 articles in comparison to fig.1 (1738).

[Answer] According to your comments, we changed “1739 articles” to “1738 articles” in comparison to fig.1. we revised the manuscript as follow. All modifications are highlighted in the manuscript

 In 3.1. Systematic review process

Subsequently, 1738 articles were removed by review of title and abstract.

6. Conclusion: page5/line 226: which optimal treatment? more concrete.

[Answer] To address your comment, we described optimal treatment concretely. we revised the manuscript as follows. All modifications are highlighted in the manuscript

In 5. Conclusion

Moreover, determination of the optimal treatment such as medical therapy, lifestyle intervention, and nutritional support for BMD loss based on various factors should help identify patients who are at risk of fracture.

Reviewer 3 Report

The authors have carried out a systematic review and meta-analysis about the effect of androgen-deprivation therapy and the mineral density of bone in patients with prostate cancer. The work is well organized and should be accepted after minor revisions.

It must be better explained the criteria to include and exclude the articles used for meta-analysis. 

There are some mistakes in english writting that should be corrected.

Author Response

1. It must be better explained the criteria to include and exclude the articles used for meta-analysis.

[Answer] To address your comment, we described include and exclude criteria concretely. we revised the manuscript as follows. All modifications are highlighted in the manuscript

 In 2.2. Trial inclusion and exclusion criteria

The inclusion criteria included a study published in English, prospective case-control design, patients with PCa or other urologic diseases, use of ADT, ADT duration of at least 6 months, follow-up period of at least 1 year, and reported values for changes in BMD of lumbar spine, femur neck, or total hip. Exclusion criteria included a cohort observational study design (no control group), use of intermittent ADT regimen, a comparison between regimens of ADT, short follow up period (< 12 months), and unable to extract outcome data. Conference and meeting abstract was also excluded.

2. There are some mistakes in english writting that should be corrected.

[Answer] To address your comment, our paper was proofread &edited by a native English speaker.